

# A spotlight on non-host resistance to plant viruses

Avanish Rai[1], Palaiyur N. Sivalingam[2] and Muthappa Senthil-Kumar[1]

[1] National Institute of Plant Genome Research, New Delhi, India
[2] ICAR-National Institute of Biotic Stress Management, Raipur, India

## ABSTRACT

Plant viruses encounter a range of host defenses including non-host resistance (NHR), leading to the arrest of virus replication and movement in plants. Viruses have limited host ranges, and adaptation to a new host is an atypical phenomenon. The entire genotypes of plant species which are imperceptive to every single isolate of a genetically variable virus species are described as non-hosts. NHR is the non-specific resistance manifested by an innately immune non-host due to pre-existing and inducible defense responses, which cannot be evaded by yet-to-be adapted plant viruses. NHR-to-plant viruses are widespread, but the phenotypic variation is often not detectable within plant species. Therefore, molecular and genetic mechanisms of NHR need to be systematically studied to enable exploitation in crop protection. This article comprehensively describes the possible mechanisms of NHR against plant viruses. Also, the previous definition of NHR to plant viruses is insufficient, and the main aim of this article is to sensitize plant pathologists to the existence of NHR to plant viruses and to highlight the need for immediate and elaborate research in this area.

## INTRODUCTION

The International Committee on Taxonomy of Viruses (ICTV) lists 1,000 different species of plant viruses, and among them, approximately 450 species are categorized as plant pathogenic viruses (*Soosaar, Burch-Smith & Dinesh-Kumar, 2005*; *King et al., 2011*). Viral diseases of crop plants threaten global agricultural production. Plant viruses are unable to penetrate the intact plant cuticle and the cell wall; therefore, they enter the plant cell by mechanical injury, are transmitted by insects or nematodes that feed on them, or by parasitic agents such as fungi. Most of the plant viruses have positive sense RNA as their genome, depend on plant cell machinery for transcription and translation, and they replicate through intermediate negative sense RNA strand by RNA dependent RNA polymerase. Similarly, negative strand RNA viruses replicate by making positive sense RNA as intermediate for multiplication and protein synthesis which depends on host machinery. Single stranded DNA viruses replicate through double stranded DNA (dsDNA) using host DNA polymerase and this dsDNA is utilized for transcription and

Corresponding author
Muthappa Senthil-Kumar,
skmuthappa@nipgr.ac.in

translation using host machinery. Thus, plant viruses modulate the intracellular milieu of the host plant, critical for the development of the viral infection, and interfere with antiviral defenses.

Viruses encode movement protein (MP), which aids in viral particle movement from cell to cell through the increased plasmodesmatal size exclusion limit (*Lefeuvre et al., 2019*). Host translational factors (eukaryotic translation initiation factors; eIFs) are involved in regulating the replication and systemic intercellular movement of viruses (*Nieto et al., 2011*; *Shopan et al., 2020*). Plants neutralize viral infection with a highly refined innate immune response, encompassing pathogen-associated molecular pattern (PAMP)-triggered immunity (PTI) and effector-triggered immunity (ETI), which resembles that of pathogenic cellular systems (*Calil & Fontes, 2017*). Viruses have small genomes, encoding roughly 6–10 proteins, hence requiring host factors to complete the infectious life cycle (*Campbell, 1996*; *Whitham & Wang, 2004*). The reliance on host factors raises the possibility that plants can develop resistance mechanisms to ameliorate the effects of viral infections during the course of evolution. This resistance mechanism of an entire plant species to a particular virus and all of its strains can be referred to as non-host resistance (NHR). This term is widely used in referring resistance to bacterial and fungal pathogens. NHR is the broad-spectrum resistance manifested by an innately immune non-host due to inducible and multilayered defense responses. In the past, inadequate progress has been made to elucidate the non-host defense mechanisms and understanding the molecular mechanisms of NHR remains poor (*Harris et al., 2020*). The main aim of this article is to provide an overview of the role of NHR through innate immune response, role of virus-vector interaction, underlying molecular mechanism of NHR and their utilization in facilitating broad-spectrum resistance to plant viruses.

## Rationale, intended audience and contribution to the field

Generally, non-host plants do not allow the virus to replicate, which phenotypically fails to show any symptoms, systemic or local. In asymptomatic plant species inoculated with certain viruses, the progeny virus are not detectable, and such plants can be considered as non-hosts. The resistance owing to the early sensing of plant viruses, which completely restricts viral multiplication in the non-host plant, can be referred to as "true NHR". However, recent evidence shows that some non-host plant species allow the multiplication of the virus till a minimal level in single or a few inoculated cells but further arrest the multiplication and cell-to-cell movement (*Sardaru et al., 2018*); this can be referred to as "apparent NHR". Molecular analysis has revealed the presence of negligible levels of virus replication, accumulation, and systemic movement in certain combinations of a non-host plant and viral pathogen. Such scenario underpin that the apparent NHR might be more common than NHR (extreme resistance or complete absence of virus replication; *Sardaru et al., 2018*; *Baruah et al., 2020*).

Non-host species mount efficient defense responses and possess diversified antiviral mechanisms that confer NHR to plant viruses. Through this brief review, we attempt to apprise plant scientists of the existence of NHR of plants to viruses and draw attention to the need to re-examine NHR to viruses from various angles. We also provide insights into

NHR and "apparent NHR", which can be exploited for providing broad-spectrum resistance. A clearer understanding of NHR will help in developing resistant crops and refining strategies and resources to achieve efficient control of plant viral diseases. Interestingly, this article delves into some critical and fundamental aspects that are often questioned by plant virologists. Except for *Baruah et al. (2020)*, that only describes introductory to the NHR, no attempt has been made to discuss this topic among plant virologists. Importantly, we discuss the role of insect vectors in non-host & thereby virus host range/resistance development. We present the summary and emphasis the need for research focus in the area of NHR. We hope this will take a step forward in discussing the topic among virologists and plant scientists in general.

## SURVEY METHODOLOGY

Several search engines including PubMed, Web of Science advanced search, Google scholar and specific journal web sites were used and search was performed based on key words specifically designed for the topic. The search keys included virus names, select plant species, non-host, virus resistance, virus co-evolution, RNA interference, silencing suppressors, effector and PAMP triggered immunity, host range, *R*-gene resistance, virus transmission, insect vectors and others. Literature was retrieved and sorted based on the relevance of the topic. The literature was mined to extract data and information from relevant articles and was used to support the hypothesis behind the review article. Various citations from these articles were referenced to obtain elaborate information. Together, the compiled information was processed by the authors to write the manuscript. Relevant concepts were incorporated based on the author's expertise in this field of research. The self and non-self recognition and topics under above key words were included. However, the literature related to animals in this regard were excluded.

### How non-host plants recognize the viruses?

The presence of nucleic acid in a cellular compartment triggers immunity in mammals and plants. Therefore, the accumulation of non-self DNA and RNA cytosolic fragments indicates infection (*Heil & Vega-Muñoz, 2019*). In mammals, single-stranded or double-stranded RNA is sensed by endosomal Toll-like receptors (TLRs), whereas dsDNA is recognized by several cytosolic receptors (*Heil & Vega-Muñoz, 2019*). In non-host plants, resistance to cellular pathogens such as fungi and bacteria is known as plant resistance to all pathovars or genetic variants of the species, while NHR against viruses should be operating at species (or even genus) level (*Baruah et al., 2020*). It is possible that direct recognition of virus by the host may not take place in the apoplast (*de Ronde, Butterbach & Kormelink, 2014*; *Sardaru et al., 2018*).

### PAMP recognition

The virus elicitors or PAMPs were recognized in a BRI1-associated kinase 1 (BAK1) manner, which believed to be the central regulator of pattern triggered immunity (PTI) (*Kørner et al., 2013*). NHR against viral pathogens may involve recognition of viral elicitors by immune signaling components which may results in host shifts or widening of host

range. In such scenario, the defense related genes may get transferred across plant species to enhance disease resistance (*Panstruga & Moscou, 2020*). Virus-derived nucleic acid might be recognized by PAMP recognition receptors, which help in the activation of the downstream NUCLEAR SHUTTLE PROTEIN-INTERACTING KINASE 1 (NIK1)-dependent antiviral signaling pathway suppressing host translation (*Teixeira et al., 2019*). In begomoviruses, virulent nuclear shuttle protein (NSP) was found to be targeted by NIK1 in the defense signaling pathway (*Zorzatto et al., 2015*).

RNA viruses interact strongly with the components of host translational machinery, such as host eIF-encoded genes (*Calil & Fontes, 2017*). In both RNA and DNA viruses, virus-derived PAMPs, dsRNA, RNA, and DNA are encoded by viral genomes that trigger antiviral PTI by activating a downstream signaling cascade through the recognition of the co-receptors BAK and SERK1 (*Teixeira et al., 2019*). The host immune sensors recognize viruses through direct and/or indirect association of viral proteins, which triggers the downstream defense signaling cascade to avert viral replication and movement within the plants (*Meier et al., 2019*). The molecular mechanism underlying NHR critically depends on the signal transduction pathways similar to other types of plant immunity. Among the set of genes, some may turn out be bottlenecks for certain pathogens on a specified plant species or under particular conditions (*Panstruga & Moscou, 2020*). The molecular studies on viral recognition suggest that the basal defense is suppressed in the compatible interaction between viral pathogens and host plants. However, the level of suppression of basal defense may be reduced in non-host plants against viral pathogens.

## Non-host, host range and resistance mechanism
### Host jump and host range

Viruses have limited host ranges, and adaptation to a new host is atypical phenomenon. Overcoming species barriers requires mutations in several viral-encoded proteins involved in cellular movement and replication. The adaptation of the virus to relatively new species is highly complex; therefore, a host jump by the virus is a very rare occurrence. Host jumps occur when pathogens encounter new hosts, followed by infection and successful multiplication in that host plant.

Host jumps occur when pathogens encounter new hosts, followed by infection and successful multiplication in that host plant. Recently, a novel coronavirus (SARS-CoV-2) crossed species barriers (its natural hosts being Chinese horseshoe bats and intermediate hosts being civet cats and raccoon dogs) to infect humans and was efficiently transmitted among humans, leading to a global pandemic (*Hoffmann et al., 2020*). Therefore, timely prediction of the emergence of viral pathogens is beneficial for both human health and agriculture.

Virus transmitting insect vectors and the three-way interaction among virus-vector-plant also plays a role in virus host range. Determinants of host range have been identified in genomes of plant viruses in various studies (*Nieto et al., 2011*; *Sardaru et al., 2018*; *Shan et al., 2018*; *Wang et al., 2020*). In spite of that, the frequency of host jumps and its consequences, such as fitness and pathogenicity are unknown for plant viruses (*Poulicard et al., 2012*). Host jumps are reported for mitoviruses, the families Rhabdoviridae and

**Table 1 Examples of determinants of NHR in plants.**

| Sl. No. | Host factors | Virus | Description | References |
|---|---|---|---|---|
| 1 | Eukaryotic translation initiation factors (eIF4E) | *Melon necrotic spot virus* | Incompatible interaction between virus RNA and eIF4E leads to *Nicotiana benthamiana* resistance against non-adapted *Melon necrotic spot virus* | *Nieto et al.* (2011) |
| 2 | ARGONAUTE2 | *Potato virus X* | RNA silencing mediated antiviral defense in Arabidopsis against PVX | *Jaubert et al.* (2011) |
| 3 | *Tm-1* | *Tomato mosaic virus* | *Tm-1* encodes a protein that binds to and inhibits the functioning of the replication proteins of *Tomato mosaic virus* | *Ishibashi et al.* (2007) |
| 4 | Constitutive PR expression 5 (CPR5) | *Brome mosaic virus* | BMV multiplies efficiently in cpr5-2 Arabidopsis mutant, which depended on the functions of RNA1 and RNA2. | *Fujisaki et al.* (2009) |
| | Viral proteins | | | |
| 5 | Helper-component proteinase (HC-Pro) | *Soybean mosaic virus* and *Clover yellow vein virus* | Although HC-Pro cistrons are functionally compatible in infection, they are not involved in determination of host range specificity | *Wang et al.* (2020) |
| 6 | P1 leader proteinase | Potyvirus | Removal of a P1 leader proteinase promotes potyvirus replication in a non-adapted host | *Shan et al.* (2018) |
| 7 | P3 viral protein | *Turnip mosaic virus* | Facilitate apparent NHR in Ethiopian mustard against *Turnip mosaic virus* | *Sardaru et al.* (2018) |

Reoviridae, and the order Bunyavirales, all of which infect both plant and insect genera (*Gilbert & Parker, 2016*). Moreover, in several plants, the mitochondrial genome contains sequences of *Mitovirus*, a genus within the Narnaviridae.

### Host cellular factors

Plant viruses move long distances through vascular systems to infect tissues (roots and young leaves), and the entire infection cycle, *i.e.*, virus replication and movement, is genetically controlled by host cellular factors (*Garcia-Ruiz, 2019*). Incompatible interactions between viral pathogens and plants are characterized by the arrest of infection processes by plants. In nature, most viral pathogens fail to establish virulence in plants without triggering a canonical immune response, and such an antiviral defense mechanism refers to NHR (*Fraser, 1990*; *Baruah et al., 2020*). The antiviral defense mechanisms in non-host involve unsuccessful viral replication, incompatible interaction between host susceptibility factors and viral proteins resulting into inhibition of translation and RNA silencing (Table 1) (*Ding, 2010*; *Nieto et al., 2011*; *Baruah et al., 2020*).

The incompatibility or the absence of cellular host factors essential for replication, multiplication, and systemic movement of viral pathogens might be one of the plant resistance mechanisms against non-adapted viruses (*Ishibashi et al., 2009*; *Nieto et al., 2011*). This shows that the resistance mechanism underlying NHR is challenging to explain, as it operates at the local or systemic level, or even in the single cell. The identification and characteristics of non-host factors that control NHR to viruses and

determinants of host range are lesser known due to the difficulty in dissecting the underlying molecular and genetic mechanisms. Accordingly, the terminology of 'true NHR' and 'apparent NHR' were defined earlier (*Baruah et al., 2020*).

The incompatible interaction between viral particles and host susceptibility proteins has been reported to be one of the factors responsible for generating NHR in tomato, a non-host for *Tobacco mild green mosaic virus* and *Pepper mild mottle virus* (belonging to the *Tobamovirus* genus) (*Ishibashi et al., 2009*). In tobacco, NHR against non-adapted *Melon necrotic spot virus* is a consequence of incompatible interaction between the virus cap-independent translational enhancer and the host eIF4E, which inhibit the protein synthesis machinery required for replication (*Nieto et al., 2011*). These results showed that the host cellular factors inhibiting the multiplication of the viral pathogen might be present in plants for conferring NHR against non-adapted viruses.

Helper component proteinase play major role in virus multiplication and determining host range specificity. For instance, *Soybean mosaic virus*-N (SMV-N) is capable of infecting cultivated soybean and wild soybean, whereas it seems to be undetectable in broad bean. In contrast, *Clover yellow vein virus* shows compatibility towards broad bean for systemic infection; whereas it is capable of infecting soybean locally (*Wang et al., 2020*). The lack of systemic infection by *Soybean mosaic virus* and *Clover yellow vein virus* in broad bean and soybean, respectively, was suggested to be due to the incompatible interaction of multifunctional helper component proteinase, HC-Pro cistrons with the respective host components (Table 1). Removal of the P1 leader protease antagonistic domain facilitates the replication of potyvirus, which strongly increases localized infection in a non-permissive *Cucumis sativus* host (*Shan et al., 2018*). To counteract non-host antiviral defense, P1 leader processing resulted in the release of a functional silencing suppressor, elucidating the adaptation and evolution of NHR to potyvirus (*Shan et al., 2018*). *Bean pod mottle virus* (BPMV) overcomes two layers of defense to move systemically in *N. benthamiana* non-hosts plants. Hence, the complete anti-BPMV defense in the non-host *N. benthamiana* consists of two layers of active defense: interference with RNA2 replication and RNA silencing (*Lin, 2013*). Previously, it has been shown that the viral P3 protein (C-terminal region) is an important NHR determinant in Ethiopian mustard against *Turnip mosaic virus* isolate JPN 1, and the resistance mechanism inferred in that study was apparent NHR (*Sardaru et al., 2018*). This is the only report available in which the term "apparent NHR" has been used. However, there are many examples available in literature to put under this category. Further, the inability of *Pea seed-borne mosaic potyvirus* (PSbMV) to infect *N. tabacum*, and of *Potato virus Y* to infect pea, could be due to inhibition of replication of viruses in mechanically inoculated cells (*Bak, Poulsen & Albrechtsen, 1998*). Moreover, it has been shown that *Cacao swollen shoot virus* (CSSV) is able to complete its life cycle in the non-host plants belonging to Brassicaceae and Solanaceae families when CSSV viral genome is stably transformed into plant genome (*Friscina et al., 2017*). This suggests that the non-host plants have all the necessary factors responsible for the replication of CSSV. The above examples clearly showed that the incompatible interactions between viral and host proteins or the lack of functional host

susceptibility factors are critical determinants that inhibit viral multiplication, which makes plants non-hosts to viral pathogens.

### RNA silencing

The molecular mechanism of NHR against virus are both active (defense) and passive (lack of susceptibility factors) in nature (*Fan & Doerner, 2012*). Antiviral silencing is the first immune response encounter by viruses while invading non-host plants.

The incompatibility between a virus and a non-host plant can be inferred by the multiple layers of antiviral defense, which consist of active suppression of viral pathogenesis and passive escape from viral multiplication (*Pallas & Garcia, 2011*).

RNA silencing is an important defense mechanism targeting viral nucleic acid and controls viral RNA detection and degradation using Dicer-like and Argonaute (AGO) proteins (*Ding & Voinnet, 2007*). For instance, *Potato virus X*, which cannot infect *Arabidopsis thaliana* is capable of infecting Dicer-like mutants in Arabidopsis when co-infected with a *Pepper ringspot virus*. Moreover, *Pepper ringspot virus* has virus suppressor of RNA silencing (VSR), which contributes to the gained infectivity of PVX. This implies that RNA-mediated antiviral silencing is responsible for NHR against PVX (*Jaubert et al., 2011*). A post-transcriptional gene silencing (PTGS) suppressor in plant veins, *Tomato bushy stunt virus* (TBSV) p19, has a crucial role in systemic invasion in the host. Moreover, *Tobacco etch virus* (TEV) helper component protease (HC-Pro) represses PTGS in tobacco but not in Arabidopsis, a non-host to TEV (*Scholthof, Scholthof & Jackson, 1995*).

### R gene-mediated resistance in NHR

*R* genes grouped into two different classes of genes: The first class belong to the genes that encode proteins which are involved in regulation of different stages of virus life cycle. The other class belong to the genes encoding NLR (Nucleotide-Binding and Leucine-Rich Repeat domain) proteins, whichplays significant role in plant-viral defense (*Meier et al., 2019*). Plants have gradually developed a complex innate immune system in response to viral pathogens and such responses are often controlled by NLR proteins. For instance, *PVR4* gene from *Capsicum annum* confers resistance against several potyviruses such as *Pepper mottle virus*, and *Pepper severe mosaic virus* suggesting that both defense response and hypersensitive response can be activated by NLR proteins (*Kim et al., 2015*).

Non-host resistance against plant viruses is conferred by several identified *R* genes, for instance, NHR may be governed by dominant/recessive genes such as *Ty-1* against *Tomato leaf curlvirus* (ToLCV) (*Verlaan et al., 2013*) or non-availability of translation initiation factor (eIF4E/eIF4G) in case of potyviruses (*Kang, Yeam & Jahn, 2005*; *Nieto et al., 2011*; *Hashimoto et al., 2016*). Also, NHR to *Brome mosaic virus* is due to *Constitutive PR expression 5* (CPR5), a protein that represses *R* gene-mediated resistance (*Fujisaki et al., 2009*). The gene encoding *R* proteins also activate ETI by interacting with RNAi suppressors protein, for instance, Tm-1 protein interacts with viral suppressors proteins such as methylase (*Ishibashi et al., 2007*; *Hashimoto et al., 2016*).

## Virus and insect vector interaction in non-host plants

Insect vectors play a significant role in expansion of geographical regions and the host ranges as evident from the case study of *Tomato chlorosis virus* (ToCV)–an emerging virus that cause economic loss of tomato (*Fiallo-Olivé & Navas-Castillo, 2019*). ToCV was first reported in Florida, USA in the mid-1990s. By 2019 this virus was reported from about 35 countries and no resistance or tolerant tomato plants are commercially available. Also, this virus infects 84 dicot plant species belonging to 25 botanical families. This kind of massive spread of virus spread is due to insect vector, whitefly. Virus also modulate hormonal signaling and insect vector perception in the infected plant as increased vector foraging and movement among plant species broaden the host ranges and also enhance the chances of virus survival (*Bera et al., 2020*). This has been reported in *Pea enation mosaic virus* (PEMV), pea aphids (*Acyrthosiphon pisum*), and pea (*Pisum sativum*), wherein oxylipin signaling pathway was identified to be necessary for aphid attraction. Moreover, the CMV 2a RNA-directed RNA polymerase of *Cucumber mosaic virus* (CMV) seems to contain PAMP activity. The activated PTI may induce production of phytoalexins that negatively affect the host and insect vector interaction, which in turn affect CMV transmission (*Carr et al., 2018*). Habitat manipulation by introducing a non-host plant through anthropogenic means has been adapted in some cases to control the spread of vector transmissible viral pathogens (*Hooks & Fereres, 2006*). Such plants are non-hosts to the virus as well as vector and act as a barrier for the non-persisted vector transmissible viruses. The virus transmitting insects spend more amount of time and energy probing on the surface of non-hosts plants compared to their host plants (*Powell, Pirone & Hardie, 1995*) and this could be a strategy used for checking viral infections. For instance, some pulse crops, which act as non-hosts to the TYLCV, are used as barrier to effectively reduce virus spread to tomato (*Campbell et al., 2017*). In host plants, viral pathogens can modify insect feeding behavior in which insects prefer to feed on infected rather healthy hosts. For instance, *Tomato spotted wilt virus* (TSWV) belonging to Bunyaviridae family changes the feeding behavior of thrips vector, *Frankliniella occidentalis* upon infection (*Stafford, Walker & Ullman, 2011*). The TSWV infected male thrips was able to spread the virus by three fold than uninfected male thrips. It has been observed that viral pathogen spread in hosts with a particular phenotype and infection stage can be changed due to vector feeding behavior. For instance, the preference of vector containing viral pathogens and vector alone largely reduce pathogen spread when diverse host species are available (*Shoemaker et al., 2019*). In addition to pathogenicity determination and modulation of vector feeding characteristic, most of the viral factors (CMV-2b, βC1 and C2) also have RNA silencing suppressor activity (*Ziegler-Graff, 2020*). Such multiple functions of viral factors make these proteins more potent in viral transmission and highlight the importance of further exploring molecular complexity present in the interactions among viral pathogens, plant non-host and insect vectors.

Hosts and insect vectors reacted to viral pathogens through the activation of immune response. RNAi, apoptosis and autophagy are the molecular mechanisms that facilitate antiviral immunity for insect vectors in response to viral infection. In the

virus-transmitting insect vectors, virus replication is restricted due to the presence of autophagic components that represent a crucial antiviral cellular response (*Chen & Wei, 2020*). For instance, TYLCV infection induces autophagy pathway in whiteflies that affect viral replication process and its exit from the insect cells (*Wang et al., 2016*; *Chen & Wei, 2020*).

In summary, the preference of insect vector foraging either on host or non-host plant is likely to depend on differences in chemical cues, plant architectural differences, and vector density. Molecular components of pathogenicity determinants—a consistent feature present in the virus-vector and virus-plant interactions. Understanding such interactions can reinforce approaches that defend plants from viruses by modulating viral pathogen transmission. In conclusion, unraveling the specific regulatory components of defense signaling and autophagy pathways in virus-transmitting vectors is needed for the understanding NHR mechanisms.

## Update on NHR-based research in plant virology

Current research findings indicate that the RNA silencing, host susceptibility factors and SA-mediated inducible defense facilitate broad-spectrum defense response in non-host plants and the same is briefly shown in the model (Fig. 1). Detailed research programs are warranted to fill the gaps in unraveling molecular mechanisms of NHR against plant viruses for better exploitation in crop protection. The use of agrochemicals against virus vectors is increasingly being discouraged from the ecological perspective. Therefore, crop protection based on naturally occurring virus resistance is an important solution to curtail crop destruction and accomplish sustainable crop yields (*Yang et al., 2014*). In organisms other than plants, NHR has been employed to control viral load; for example, the abundance of the virus PgV-07T, which infects the marine alga *Phaeocystis globosa*, was significantly reduced by using non-host marine organisms such as crabs, cockles, oysters, sea squirts, and sponges (*Welsh et al., 2020*).

Understanding of the underlying mechanisms involving variability, shifts, and evolution of host range are needed. In fact, NHR in plants depends on the affinity between plant cellular proteins or susceptibility factors (tm-1, eIFs, and AGO2) and viral proteins (*Ishibashi et al., 2009*; *Nieto et al., 2011*; *Jaubert et al., 2011*). Non-hosts serve as a buffer in ecology and are necessary for balanced host occupancy and spread. Since these host factors are relatively conserved, the capacity of a given viral pathogen to overcome barriers of infection in various host plants decreases with the degree of divergence between these host factors and, consequently, with the divergence time between these plants. In contrast to other pathogens, viruses have host ranges, which could be functionally determined by arthropod vectors (*Gilbert et al., 2012*).

The host range of viruses is a significant determinant of disease emergence and can also be used in designing strategies for disease control. A recent study unraveled the diversity and frequency of changes in non-host/host ranges over the evolutionary time scales of potyviruses and further determined the distribution of these changes among potyvirus and plant diversity (*Moury & Desbiez, 2020*). The study suggested that NHR to plant viruses results from a series of inducible defenses in non-host plants that viruses cannot overcome

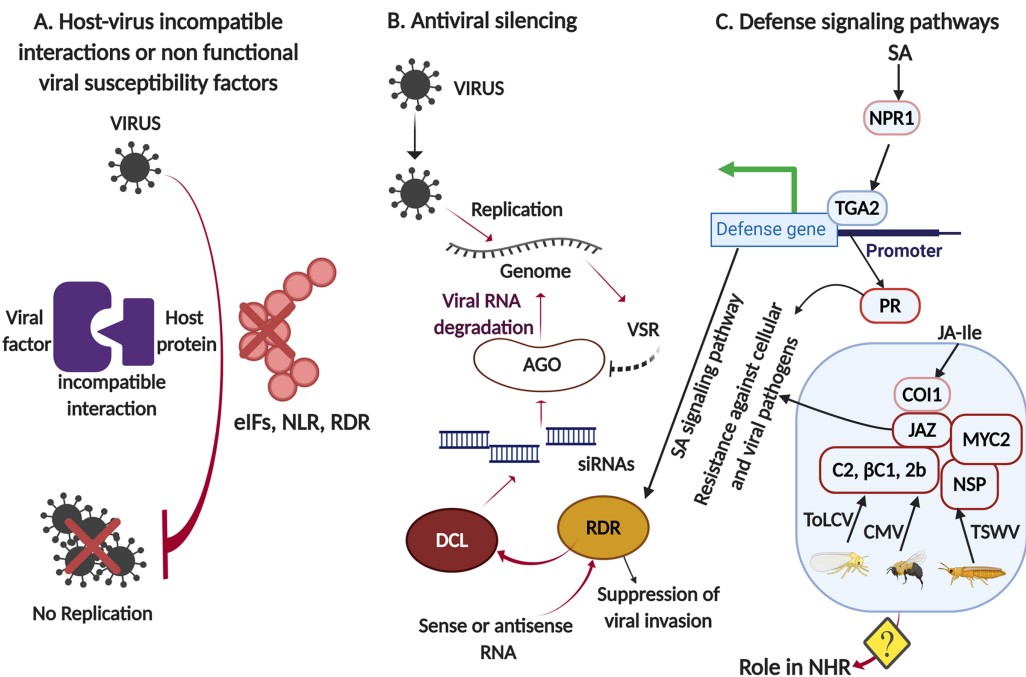

**Figure 1 A generalized overview of the potential molecular mechanisms of non-host resistance to viruses.** (A) The incompatible interaction between viral factors and host proteins or the members of non-functional viral susceptibility proteins, such as eIF, translation initiation factor; NLR, nucleotide-binding domain leucine-rich repeat; and RDR, RNA-dependent RNA polymerase, can provide resistance to viruses by inhibiting viral multiplication. (B) A simplified mechanism showing antiviral silencing in which RDR creates an amplification loop in the process of RNA degradation, mediated by the dsRNA-specific endo-ribonuclease Dicer-like (DCL) family of proteins. DCLs detect and process dsRNA into small interfering RNAs (siRNAs). Argonaute (AGO) proteins play a major role in RNA silencing which further functions distinctively in transcriptional and post-transcriptional gene silencing. Viral suppressors for RNA silencing (VSR) repress RNA silencing, having evolved to be encoded by plant viruses to neutralize resistance mechanisms by antiviral silencing in plants. (C) Salicylic acid (SA) significantly induces the expression of RDR in the presence of TMV infections, which shows a critical link between RNA silencing and the nonexpressor of PR-1 (NPR1)-dependent SA signaling pathway. Abbreviations: COI, CORONATINE INSENSITIVE-1; JAZ, jasmonate ZIM domain; NSP, non-structural proteins.

by faster co-evolution. In addition, host adaptation to vertically passaged viruses is traded-off against diminished resistance to the non-evolved viruses, and this mode of spread and host–virus co-evolution is a major factor in determining the evolution of virulence (*Pagan et al., 2014*). NHR is robust and multigenic trait involving layers of distinct processes (*Maule, Caranta & Boulton, 2007*), which could be exploited for the development of virus-resistant plants. To achieve this, detailed molecular analyses are required for understanding multilayered constitutive and inducible defense mechanisms involved in NHR to plant viruses. Viruses have the ability to evolve rapidly to counter host resistance; therefore, it is necessary to understand determinants that influence the robustness of resistance traits to maximize their endurance. Wild relatives of cultivated plants may represent rich resources of resistance genes for elucidating the molecular mechanism of NHR to plant viruses and understanding the robustness of resistance traits (*Ishibashi et al., 2007*). For example, Tm-1 protein was identified from *Solanum*

*habrochaites*, a wild relative of tomato, and it cannot bind to TMV replication proteins or inhibit its multiplication (*Ishibashi et al., 2007, 2009*). These opportunities are exploitable for generating novel resistance, as well as for providing an understanding of broad-spectrum NHR to plant viruses through clustered regularly interspaced short palindromic repeats (CRISPR-Cas9) targeted genome engineering, which can be utilized to integrate dominant resistance traits at a single locus for the development of virus-resistant plants and crop improvement (*Zhao et al., 2020*). Several viral proteins have the capability to suppress JA-regulated gene expression, for instance, begomovirus satellite-βC1, *Cucumber mosaic virus* 2b, and TuMV HC Pro proteins (*Wu & Ye, 2020*). Moreover, the activation of SA signaling has proven effective against different plant viruses, *e.g.*, SA-inducible *RdRP* from *Nicotiana tabacum* is required for defense against members of the *Tobamovirus* genus but not CMV or PVX (*Yang et al., 2004*). In another study it has been shown that the reconstituting Rdr1 activity in *N. benthamiana* provided protection against extreme weather conditions (*Bally et al., 2015*). Moreover, knocking out the functional allele of Rdr1 in a wild strain turned it into hypersusceptible TMV-U1 (*Bally et al., 2015*). The host cellular factors inhibiting the multiplication of the viral pathogen might be present in plants for conferring NHR against non-adapted viruses (Fig. 1). Additionally, recognition of viral proteins by plant susceptibility factors may induce canonical defense pathways involving SA and JA signaling, which may restrict the growth of non-adapted viral pathogens. However, the role of SA/JA signaling pathways conferring NHR to plant viruses is poorly understood, and further molecular evidence is needed to corroborate the role of phytohormones in NHR against viruses.

# CONCLUSIONS

NHR against plant viruses differ from resistance to other plant pathogens. Lack of host factors, incompatible interaction of host and viral factors, restriction in movement and spread, RNAi silencing, to some extent *R* gene mediated, and other mechanisms are part of NHR for plant viruses. The evolutionary interaction between virus, vector and plants prepares a plant to NHR or host jump. Understanding NHR against plant viruses and classifications are at the infancy stage and need much more systematic study. The apparent NHR looks promising for use in a resistance enhancement program, which can utilize recent genome editing tools like CRISPR on recessive host factors. We further need to understand the mechanistic diversity of antiviral defense in non-host plants and this article will help in building novel research strategies and resources for the utilization of NHR to achieve adequate control of viral infections.

## Funding

The non-host resistance project is funded by an ICAR-National Agricultural Science Fund grant (F. No. NASF/ABP-7021/2018-19/252). The funders had no role in study design, data collection and analysis, decision to publish, or preparation of the manuscript.

## Grant Disclosures

The following grant information was disclosed by the authors:

ICAR-National Agricultural Science: NASF/ABP-7021/2018-19/252.

## Competing Interests

The authors declare that they have no competing interests.

## Author Contributions

- Avanish Rai performed the experiments, analyzed the data, prepared figures and/or tables, authored or reviewed drafts of the paper, and approved the final draft.
- Palaiyur N. Sivalingam analyzed the data, authored or reviewed drafts of the paper, and approved the final draft.
- Muthappa Senthil-Kumar conceived and designed the experiments, authored or reviewed drafts of the paper, and approved the final draft.

## Data Availability

This article does not have raw data. All information is presented in the text.

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
