# Peer review of "A spotlight on non-host resistance to plant viruses"

_PeerJ, doi:10.7717/peerj.12996_

## Round 0.1 · original submission · Minor Revisions

Dear Authors

Your manuscript "A spotlight on non-host resistance to plant viruses" has been reviewed by two experts and their comments are attached below.

Both of the reviewers agreed to accept your manuscript after revision.
Please reply to all the points raised by the reviewers when you resubmit your revised version of the manuscript.

Thanks ahead for your cooperation.

Sincerely,

Reviewer 1 ·

Basic reporting

no comments

Experimental design

no comments

Validity of the findings

no comments

Additional comments

The authors described about the possible mechanisms of nonhost resistance against plant viruses in multiple aspects. Authors suggested suppression of viral replication could be due to host factor – virus incompatibility, RNAi mechanisms in terms of adaptation of virus on host / nonhost machineries. Furthermore, the authors also introduced several researches about interactions between host plant – insect vector - virus which may contribute to host range of viruses.
The authors enumerated various researches about NHR against plant viruses including the recent cases (2019 ~ 2020). Though the first half of the manuscript is conceptually not much distinguished from the previous review articles (such as Baruah et al., 2020, Panstruga and Moscou, 2020), this manuscript introduce more current researches about NHR against plant viruses.
For the other half of the manuscript, virus and insect vector interaction part provides intriguing aspects. However, this part should be more organized (to be flowing well) and specific to support the statements.
This manuscript provides useful information about current state of NHR against plant viruses. However, some parts are not easily understood, so I would like to offer some suggestions to improve them to be published.

Major points
1. Some parts are not enough specifically (vaguely) described about references/previous researches to support statements. For example, Line 203 – 205, 262 – 263, 269 – 274, 286 – 288, 288 – 291, 300 – 303, etc (other detailed comments are attached in Minor points).
2. Authors defined the ‘true’ and ‘apparent’ NHR, but in this manuscript, only one case of previous research (Line 214 – 217) was designated as apparent NHR.
3. I suggest to add table containing the information about which referenced researches are true/apparent NHR, or which specific factors are (+ how) contributed to those NHR according to the author’s classification (RNA silencing, host target incompatibility etc).

Minor points
Line 21, does NHR is another mechanism compared to host defenses?
Line 25 & 64, does NHR is non-specific resistance? Authors described NHR as associated with recognition based or non-specific resistance in line 156 ~159.
Line 67 – 70, cannot easily understand this sentence
Line 125, NHR plants => nonhost plants
Line 147, no references of ‘various studies’
Line 163 – 166, description about arms race between plant RNA silencing and virus suppressor seems not well matched with sub-heading: Active and passive mechanisms.
Line 188 – 190 and 192 – 195, referenced same article, divided paragraph seems not necessary and confusing.
Line 242 – 248, authors never described what’s R gene in manuscript, clarify its definition, and R gene-mediated resistance section should be more described in detail. This section only composed of several examples without clear/distinct definition of concept and conclusive descriptions.
Line 255 – 260, redundantly re-defined NHR and described about its multi-layered nature.
Line 250 – 267, authors only described about cytosolic receptors and NLRs in this section but claim as PTI and ETI may function for recognizing viruses in plants. Maybe Line 269 – 280 (section described about PTI) could be better to be integrated with Line 250 – 267.
Line 307 – 312, no reference
Figure 1. These is no ‘D’ section in the image

Reviewer 2 ·

Basic reporting

The authors have reviewed nonhost resistance mechanism to plant viruses. Different to fungi or bacterial pathogen, plant viruses have a specific feature to infect plants. In this point, introduction needs to be revised to include plant virus infection process such as virus recognition, replication, and how to cause disease. The paragraph to summarize virus infection process could make the readers to understand the manuscript more clearly.

Experimental design

In the section of 'Rationale, intended audience and contribution to the field', the first paragraph(line 80-93) and the second paragraph(line 95-101) are overlapped with the next section 'NHR:the terminology and its usage'. I think that those sections need to be merged into one section.

I think that the section of 'How non-host plants recognize the viruses' needs to be moved next to introduction section. This section seems to mention host resistance mechanism against plant virus, which suggest the possibility to occur similar mode of action in NHR. Some NLR proteins such as ty-2 and pvr4 are known as resistance genes against virus. As the authors emphasize the improvement of crop resistance to virus, NLR genes against virus needs to be mentioned.

Validity of the findings

No comment

Additional comments

The authors have reviewed nonhost resistance to plant viruses in several aspects including host range, host factors, and insect vectors. The manuscript is well-written and provides insight to durable resistance against viral pathogen. If the manuscript is re-organized, it could be more easy to understand.

---

## Round 0.2 · Minor Revisions

Dear Author

Your resubmitted manuscript has been reviewed by the original reviewers.

Both of the reviewers agreed that the new version is improved.
However, they still have minor comments for improving your manuscript.

Please revise it for the final version.

Thanks for your cooperation.


Sincerely,

Reviewer 1 ·

Basic reporting

The authors revised manuscript as suggested.

The main body of manuscript is now divided into four parts

1. How the plants recognize viruses
2. Possible mechanisms of NHR against virus in plants
3. Virus and insect vector interaction in non-host plants
4. Update on NHR-based research in plant virology

In the first part, authors discuss only about PAMP recognition, but start with a sentence as 'NLR plays in significant role in plant-viral defense', the first sentence seems not matched for this part.

In the third part, the authors enumerates a bunch of case studies but its hard to draw a conclusion, I hope the authors add conclusive paragraph summary the insight from those case studies at the end of section.

Experimental design

no comments

Validity of the findings

no comments

Additional comments

Line 162 - only this subheading contain (:) at the end of text
Line 175 - No period (.) at the end of sentence

Reviewer 2 ·

Basic reporting

The manuscript has been revised according to the suggestions and comments of the reviewers. This manuscript could help readers to understand plant nonhost resistance mechanism against viral pathogen. Some minor points could be revised to improve the manuscript.

1. Line 141. The intracellular receptor ~~ : This sentence doesn't fit in the paragraph 'PAMP recognition'. it needs to move to (v) R-gene mediated resistance in NHR.

2. The paragraph of (ii) Active and passive mechanisms (Line 212 - Line 220) : in this paragraph, the author mentioned 'antiviral silencing' briefly and it is not enough to represent active and passive mechanism. The section seems like 'introtduction' of other section. Also, RNA silencing has been well written in the next section of '(iv) RNA silencing' I think this paragraph could be deleted and merged into other section.

3. Line 246. In revised manuscript, there is no description of true NHR and apparent NHR. The sentence of 'Accordingly, the terminology of 'true NHR' and apparent NHR' were defined earlier.' does not fit in well with this paragraph. I think that it needs to be explained more for readers.

4. Line 298. A. thaliana : Arabidopsis thaliana (appeared first in the manuscript)

Experimental design

no comment

Validity of the findings

no comment

Additional comments

no comment

---

## Round 0.3 · accepted · Accept

The current version of the manuscript could be accepted as it is.